# Breast cancer incidence and predictions (Monastir, Tunisia: 2002–2030): A registry-based study

Imen Zemni[1,2,3]*, Meriem Kacem[1,2,3], Wafa Dhouib[1,2,3], Cyrine Bennasrallah[1,2,3], Rim Hadhri[4], Hela Abroug[1,2,3], Manel Ben Fredj[1,2,3], Moncef Mokni[5,6], Ines Bouanene[1,2], Asma Sriha Belguith[1,2,3]

1 Department of Epidemiology and Preventive Medicine, Fattouma Bourguiba University Hospital, University of Monastir, Monastir, Tunisia, 2 Faculty of Medicine of Monastir, Department of Epidemiology, University of Monastir, Monastir, Tunisia, 3 Technology and Medical Imaging Research Laboratory—LTIM—LR12ES06, University of Monastir, Monastir, Tunisia, 4 Department of Pathology, Fattouma Bourguiba University Hospital, University of Monastir, Monastir, Tunisia, 5 Faculty of Medicine of Sousse, Department of Pathology, Farhat Hached University Hospital, University of Sousse, Sousse, Tunisia, 6 Cancer Register of the Center, Sousse, Tunisia

* imen11zemni@gmail.com

**Data Availability Statement:** All relevant data are within the article and its Supporting Information files.

## Abstract

### Introduction

Breast cancer is a major public health problem worldwide. It is the leading cause of cancer deaths in females. In developing countries like Tunisia, the frequency of this cancer is still growing. The aim of this study was to determine the crude and standardized incidence rates, trends and predictions until 2030 of breast cancer incidence rates in a Tunisian governorate.

### Methods

This is a descriptive study including all female patients diagnosed with breast cancer in Monastir between 2002 and 2013. The data were collected from the cancer register of the center. Tumors were coded according to the 10th version of international classification of disease (ICD-10). Trends and predictions until 2030 were calculated using Poisson linear regression.

### Results

A total of 1028 cases of female breast cancer were recorded. The median age of patients was 49 years (IQR: 41–59 years) with a minimum of 16 years and a maximum of 93 years. The age-standardized incidence rate (ASR) was of 39.12 per 100000 inhabitants. It increased significantly between 2002 and 2013 with APC of 8.4% (95% CI: 4.9; 11.9). Prediction until 2030 showed that ASR would reach 108.77 (95% CI: 57.13–209.10) per 100000 inhabitants.

**Funding:** The author(s) received no specific funding for this work.

**Competing interests:** The authors have declared that no competing interests exist.

**Abbreviations:** GCO, Global Cancer Observatory; ICD-10, International Classification of Diseases 10; CIR, Crude Incidence Rate; ASR, Age Standardized Rate; APC, Annual Percentage Change; IQR, Interquartile Range; SEER, Surveillance, Epidemiology, and End Results; HDI, Human Development Index; US, United States; LMICs, Low- and Middle-income Countries.

## Conclusion

The incidence and the chronological trends of breast cancer highlighted that this disease is of a serious concern in Tunisia. Strengthening preventive measures is a primary step to restrain its burden.

## Introduction

Breast cancer is a major public health problem worldwide. It is the most common cancer in women. Among females' malignant neoplasms, it represents the leading cause of preventable cancer deaths in the world [1]. The Global Cancer Observatory (GCO) database estimates that there were 2 261 419 new female breast cancer cases and 684 996 related deaths in 2020 [2]. Moreover, morbidities generated by this cancer as well as the cost of therapies used make it a heavy health, social and economic burden.

It is commonly recognized that high incidence rates were observed in developed countries, whereas the mortality rate was high in low-middle income countries [3,4]. However, the rapid pace of urbanization in developing countries in the last decades, has created changes in many disease patterns. The incidence of breast cancer has been steadily increasing and is currently facing unexpected challenges in management [5]. Moreover in these countries and specifically in North African countries breast cancer is characterized by an earlier age, an advanced stage at presentation and more aggressive subtypes compared to Western countries [6].

Hence, preventing this cancer is among the public health priorities. Assessing cancer patterns and trends is essential for setting the health care priorities, identifying targets for intervention as well as guiding further research [7,8]. Thus, the availability of cancer data is a key element in setting up a program to fight this disease. Thus, implementing registers which provide reliable information on the cancer profile is essential. As such, in 1966, the International Association of Cancer Registries (IACR) was created [9].

In Tunisia, breast cancer is the first female cancer and represents the leading cause of cancer-related deaths (19.7%) [10]. According to the GCO, the number of Tunisian new cases of breast cancer in 2020 was of 3092 (34.5% of all female cancers) [11]. Three Tunisian cancer registers were created in 1987 (North, Center, and South). However, data issued from these registers are not being updated. The latest estimations of 2020 available in the GCO were generated from the Northern registry (2008–2010) and the central registry (2003–2007) [2]. The most recent studies describing the actual field data on breast cancer in Tunisia are based on these old data. All other recent publications on breast cancer epidemiology in Tunisia are based solely on estimates from the Global cancer observatory.

The aim of this work was to determine the crude and standardized incidence rates, trends over a period of 12 years (2002–2013) and predictions until 2030 of breast cancer incidence rates in Monastir governorate based on data from a population based cancer registry.

## Methods

### Study design

This is a descriptive study including all female patients diagnosed with breast cancer in Monastir between 2002 and 2013.

### Setting

Monastir Governorate is one of the twenty-four governorates of Tunisia. It is located on the north-eastern coast of Tunisia. It is an industrial governorate with several industries including

textile, brick, and soap factories. It covers an area of 1019 km2 (393 m$^2$) and is divided into thirteen delegations. In 2014, the female population of Monastir counted for 274 613 persons and represented 5% of the female Tunisian population and 22% of the female population of the center region of the country composed of six governorates: Sousse, Monastir, Mahdia, Kairouan, Kasserine and Sidi Bouzid. Data about cancer incidence in Monastir is recorded continuously in a population-based register (the center registry of cancer) which is a regional registry that records cancer data for the above cited six governorates. However, latest data published from this registry dates from 2003–2007 [2].

## Participants

We analyzed all breast cancer new cases residents in Monastir Governorate from January 2002 to December 2013. Male breast cancer cases were excluded from our analysis.

## Data collection

Female breast cancer new cases in Monastir were selected from the cancer register of the center between 2002 and 2013 according to the 10[th] version of international classification of diseases (ICD-10) [12]. Verification of data conformity between cases recorded in the center registry of cancer and cases recorded in Monastir health care centers was carried out by the team of the Department of Epidemiology and Preventive Medicine of Monastir University Hospital.

## Variables

Data included variables related to age, date of first diagnosis and residential address.

## Statistical analysis

Data were verified and analyzed using IBM SPSS Statistics for Windows, Version 20.0. Armonk, NY: IBM Corp and Microsoft Excel. The crude incidence rate (CIR) of breast cancer was calculated based on Tunisian National Institute of Statistics data and was expressed per 100,000 person-years [13]. As we have included only females with new breast cancer in our study, we used female population as denominator. The age standardized incidence rate (ASR) per 100,000 person-years was calculated using the world standard population according to the World Health Organization statement of 2013 [14]. To test trends of the CIR we have calculated the Annual Percentage Change (APC) using Join point regression program. We estimated 2030 breast cancer incidence predictions using the Age Period Cohort analysis based on Poisson log linear regression. A p-value < 0.05 was considered statistically significant.

## Ethical considerations

The study was conducted according to ethical standards collections. To maintain the principle of confidentiality, the data used were anonymized. Ethics approval for the conduct of the research was gained from the Ethics Committee of the faculty of medicine of Monastir (Reference Number: IORG0009738N˚101/OMB0990-0279).

# Results

## Participants

A total of 1028 cases of female breast cancer were collected which represented 27.4% of all female cancer cases recorded in Monastir Governorate from January 2002 to December 2013.

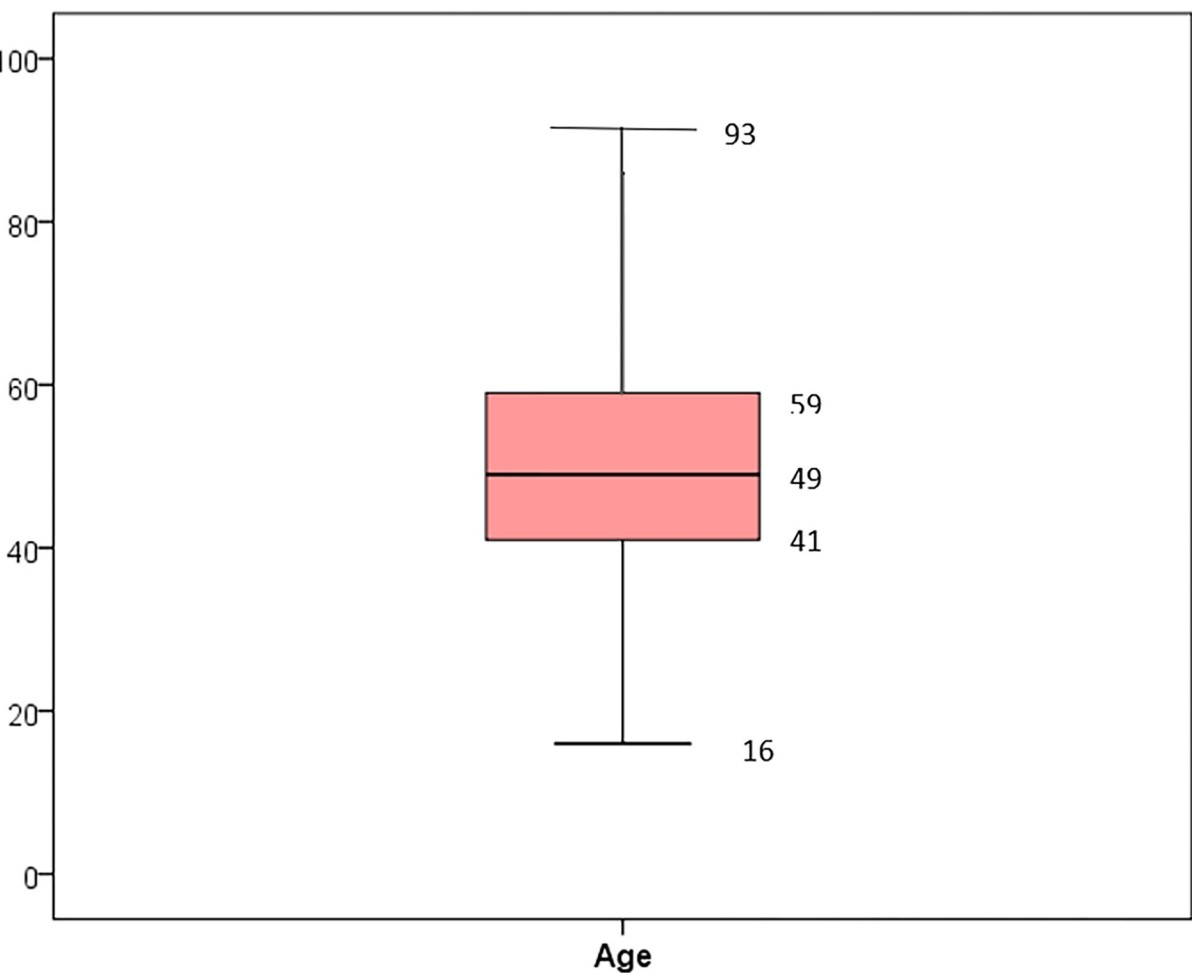

**Fig 1. Age of diagnosis of breast cancer (Monastir, 2002–2013).**

The median age of patients was of 49 years (IQR: 41–59 years) with a minimum of 16 years and a maximum of 93 years (Fig 1). About 10.5% and 23% of breast cancer cases occurred in women under 35 years old and under 40 years respectively (Fig 2).

## Crude and standardized incidence rates of breast cancer

The crude incidence rate was of 34,25 per 100000 inhabitants. The highest CIR was noted for patients aged between 40 and 59 years (101.7 per 100000 inhabitants) (Fig 3). The age-standardized incidence rate (ASR) was of 39.12 per 100000 inhabitants.

## Trends and predictions

Breast cancer incidence increased significantly between 2002 and 2013 with APC of 8% (95% CI: 4.6; 11.6) for crude incidence rate and 8.4% (95% CI: 4.9; 11.9) for Standardized incidence rate (Fig 4). With regard to age group, trends for patients aged more than 40 years increased significantly during the study period (Table 1).

Prediction until 2030 showed that CIR would reach 123.76 (95% CI: 64.76–238.96) and ASR would reach 108.77 (95% CI: 57.13–209.10) per 100000 inhabitants (Table 2). Predictions according age group are shown in Table 3.

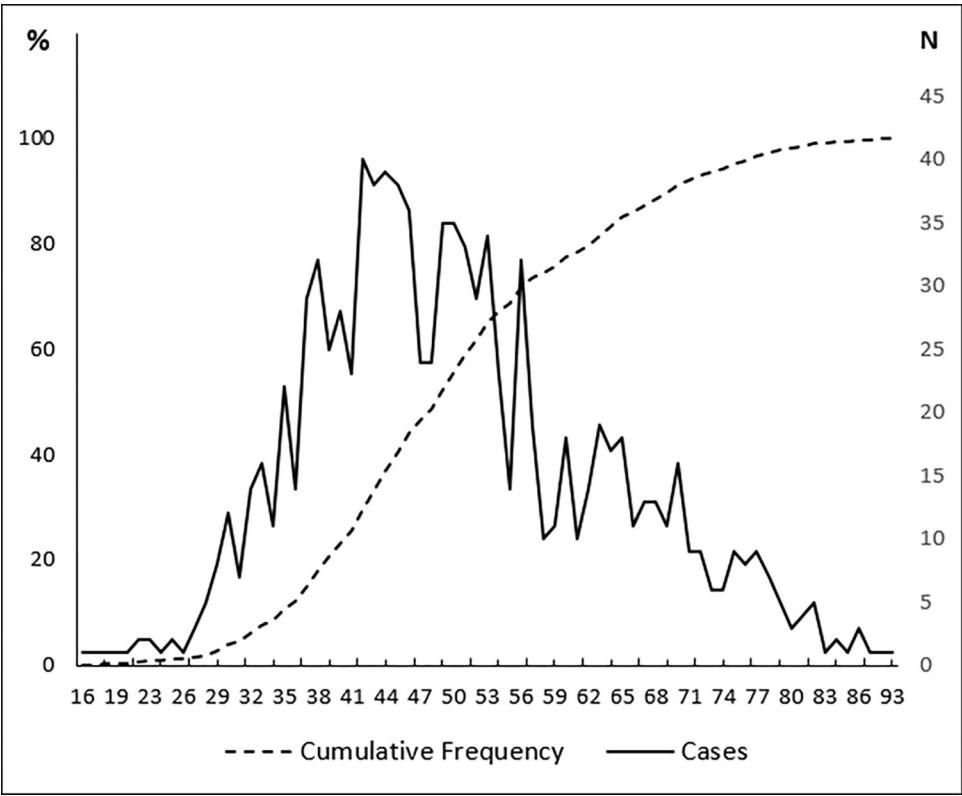

**Fig 2. Distribution of female breast cancer new cases according to the age of diagnosis (Monastir, 2002–2013).**

## Discussion

Disparities between countries in terms of the epidemiology and the management of breast cancer are well known. However, it remains a rising public health problem globally. The present study described the epidemiology and trends of breast cancer in Monastir Governorate from 2002 to 2013 with projection to 2030.

The ASR of female breast cancer in Monastir was of 39.12 per 100000 inhabitants. Our results showed a higher incidence rate than that reported in the Center of Tunisia between 1993-2007(ASR of 29.2 per 100,000) and the North of Tunisia between 2007–2009 (ASR:35.1/ 100 000) [15,16]. In addition, compared to the GCO 2012 estimations, our findings were higher than the rate observed previously in the global Tunisian population (ASR: 31.8 per 100 000). These data raise the hypothesis of a higher breast cancer incidence rate in Monastir in comparison to the national rate. This may be partly attributable to the industrialized feature of the region with higher air pollution levels and an urbanized lifestyle. Comparison with results from other neighboring countries showed higher rates in Northern Africa and Western Asia (43.2 and 42.8 per 100 000 respectively) [17]. When comparing with statistics in developed countries, much higher incidence rates were observed in very high Human Development Index (HDI) countries led by the Netherlands (117.2/100,000) [2,18,19]. In France, breast cancer remains the most frequent cancer in women with an ASR of 92.2 per 100,000 in 2007 [20]. In 2010, based on cases diagnosed from 17 surveillance, epidemiology, and end results (SEER) geographic areas, the United States reported that the ASR of female breast cancer was 126.02 per 100,000 women [21]. In Russia the ASR was of 45.6 per 100 000 between 2009–2013 [7] However, the incidence rate reported in our study remains superior to rates reported from

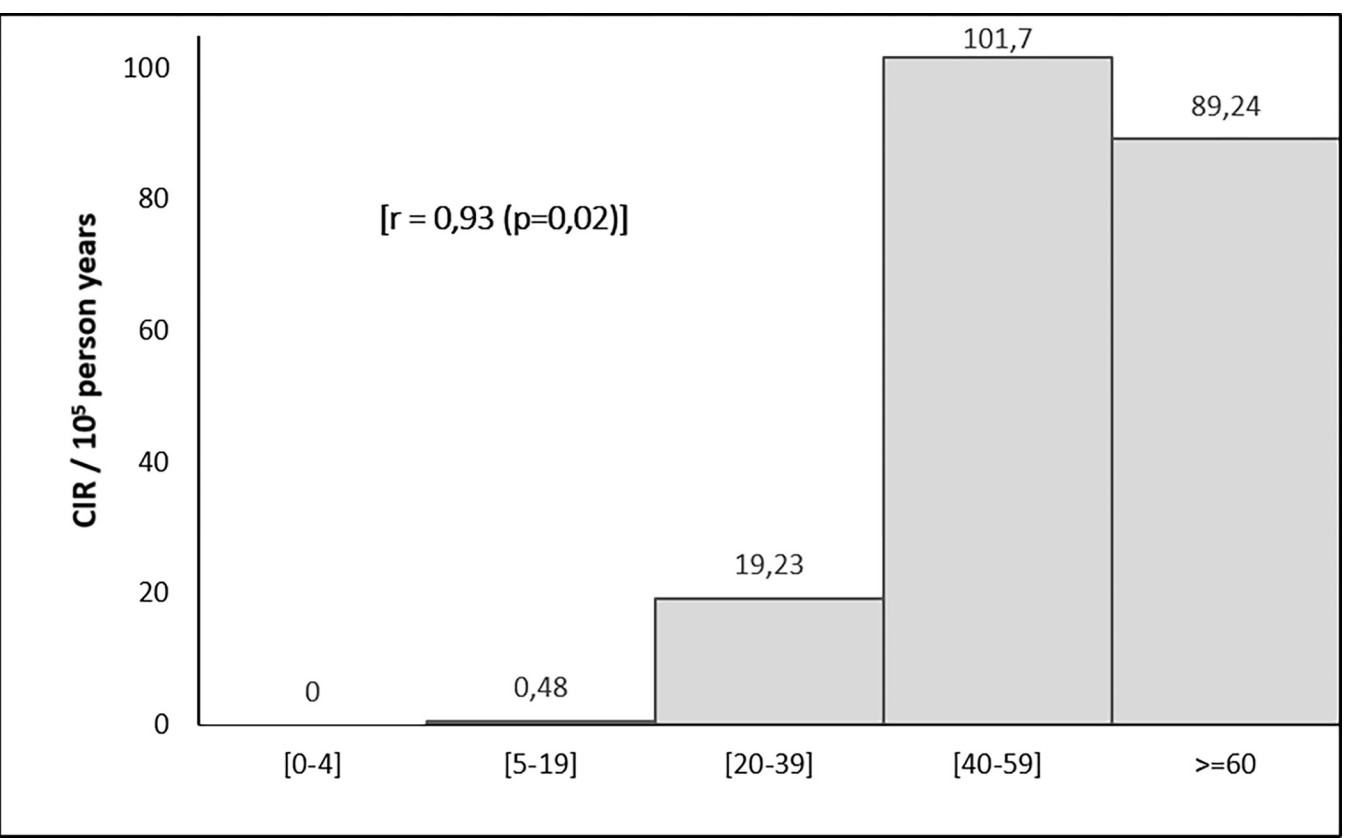

**Fig 3. Crude incidence rate of breast cancer by age group (Monastir, 2002–2013).**

South-central Asia and Eastern Asia, (ASR: 27–28.2) [19]. These differences are in accordance with the literature. Indeed, breast cancer is most frequent among developed countries than developing ones, whereas the mortality rate is higher in low/medium HDI countries [7,22,23]. This can be partially explained by a more protective risk profile in low- and middle-income countries especially North-Africa than in Western countries. The efficiency of the screening programs in Western countries may also explain this difference [6].

Nevertheless, female breast cancer had a significant increasing trend with an APC of 8% between 2002–2013 and ASR was predicted to be increased to 108.77 (95% CI: 57.13–209.10) per 100000 inhabitants.

These results are consistent with previous Tunisian reports on breast cancer. Indeed, from 1993–2007 a positive trend was noted (+2.5%) in Central Tunisia and Northern Tunisia (an APC of 1.5% between 1994–2009) [15,16]. Moreover according to the GCO estimations the incidence of breast cancer would continue to increase in Tunisia and would reach 370 per 100000 inhabitants in 2030 [16,24].

However, according to this study a decrease was noted in 2012 which can be explained mainly by the lower quality of notification and the screening program during this period marked by the Tunisian revolution and the political and economic instability. Cases which were not registered during 2012 were notified in 2013. That is why, a sharp increase in the incidence rate during 2013 was recorded.

Worldwide, incidence rates of breast cancer are rising fast in transitioning countries. According to Lima SM et al, Middle East North Africa had the largest per-year increase in

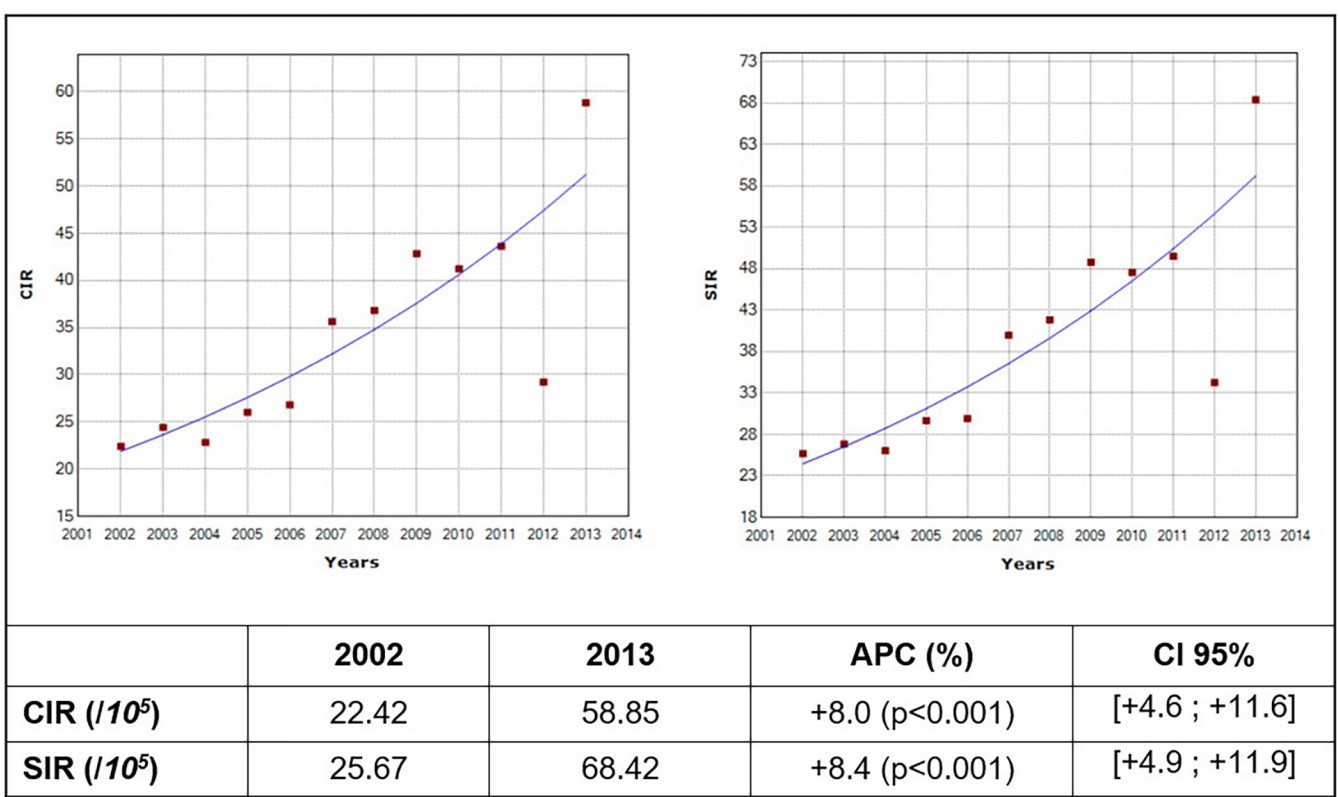

| | 2002 | 2013 | APC (%) | CI 95% |
|---|---|---|---|---|
| CIR (/10⁵) | 22.42 | 58.85 | +8.0 (p<0.001) | [+4.6 ; +11.6] |
| SIR (/10⁵) | 25.67 | 68.42 | +8.4 (p<0.001) | [+4.9 ; +11.9] |

**Fig 4. The annual percentage change (APC) of breast cancer crude and standardized incidence rates (Monastir, 2002–2013).**

overall incidence between 1990 and 2017 (APC = 2.38, 95% CI = 2.29, 2.47) [25]. Sung H et al reported that some of the most rapid increases are occurring in sub-Saharan Africa with an increase rate >5% per year in Malawi (Blantyre), Nigeria (Ibadan), and the Seychelles and of 3% to 4% per year in South Africa between the mid-1990s and the mid-2010s. Rates are rising also in transitioning Asian countries as well as in some high-income Asian countries (Japan and the Republic of Korea) [26,27] where rates are historically low but expected to continue to increase [28]. This increase can be linked to a number of risk factors such as changes in lifestyle and reproductive factors related to growing economies and the increase in the proportion of women in the industrial workforce (Diet, greater levels of excess body weight, physical inactivity, postponement of childbearing, having fewer children and lower breastfeeding. . .) [29] and have resulted in a convergence toward the risk factor profile of western countries and narrowing international gaps in breast cancer morbidity [26]. By contrast, several reports documented

**Table 1. The annual percentage change (APC) of female breast cancer crude incidence rate by age group (Monastir, 2002–2013).**

| Age class (Years) | CIR (/10⁵)* | | APC (%) | CI 95% |
|---|---|---|---|---|
| | 2002 | 2013 | | |
| 20–39 | 11.42 | 25.13 | +2.7 (p = 0.3) | [-3.7; +9.6] |
| 40–59 | 66.49 | 169.44 | +8.7 (p<0.001) | [+5.8; +12.2] |
| ≥ 60 | 59.73 | 191.99 | +10.9 (p<0.001) | [+5.2; +16.9] |

*CIR: Crude Incidence Rate (/10⁵ person-years).

**Table 2. Projection of female breast cancer crude and standardized incidence rates per $10^5$ person-years until to 2030 in the Monastir region.**

| Year | N | 95% CI | | CIR* | 95% CI | | SIR** | 95% CI | |
|---|---|---|---|---|---|---|---|---|---|
| | | Lower limit | Upper limit | | Lower limit | Upper limit | | Lower limit | Upper limit |
| 2016 | 158 | 123.37 | 203.09 | 55.16 | 42.89 | 70.61 | 56.7 | 44.13 | 72.63 |
| 2017 | 171 | 129.69 | 225.44 | 58.3 | 44.13 | 76.71 | 59.44 | 44.98 | 78.19 |
| 2018 | 185 | 136.33 | 250.5 | 61.68 | 45.42 | 83.47 | 62.24 | 45.82 | 84.21 |
| 2019 | 200 | 143.34 | 278.57 | 65.42 | 46.86 | 91.08 | 65.44 | 46.84 | 91.12 |
| 2020 | 216 | 150.75 | 310.01 | 69.24 | 48.23 | 99.19 | 68.16 | 47.45 | 97.65 |
| 2021 | 234 | 158.56 | 345.24 | 73.33 | 49.66 | 108.14 | 71.54 | 48.42 | 105.51 |
| 2022 | 253 | 166.84 | 384.71 | 77.68 | 51.15 | 117.95 | 74.93 | 49.32 | 113.78 |
| 2023 | 274 | 175.59 | 428.96 | 82.29 | 52.68 | 128.7 | 78.49 | 50.22 | 122.72 |
| 2024 | 297 | 184.84 | 478.58 | 87.21 | 54.25 | 140.47 | 82.23 | 51.15 | 132.38 |
| 2025 | 322 | 194.64 | 534.22 | 92.42 | 55.88 | 153.37 | 86.14 | 52.09 | 142.82 |
| 2026 | 349 | 205.01 | 596.66 | 97.96 | 57.55 | 167.5 | 90.25 | 53.06 | 154.1 |
| 2027 | 378 | 215.99 | 666.74 | 103.84 | 59.28 | 182.99 | 94.55 | 54.04 | 166.3 |
| 2028 | 410 | 227.6 | 745.44 | 110.08 | 61.05 | 199.96 | 99.07 | 55.04 | 179.47 |
| 2029 | 445 | 239.91 | 833.83 | 116.72 | 62.88 | 218.56 | 103.81 | 56.08 | 193.71 |
| 2030 | 483 | 252.92 | 933.16 | 123.76 | 64.76 | 238.96 | 108.77 | 57.13 | 209.1 |

*CIR: Crude Incidence Rate (/$10^5$ person-years).

**SIR: Standardized Incidence Rate (/$10^5$ person-years).

recent declines in the incidence of breast cancer in the US and throughout developed countries [25,30,31]. In fact, during the early 2000s, incidence dropped or stabilized [32] largely due to a reduction in the use of menopausal hormone therapy and a plateau in screening participation [33,34]. But since 2007, a slow upturn in incidence rates has been notified in the United States (<0.5% annually) [35] and in many other European countries [27]. This increase was limited to estrogen receptor-positive cancer which was attributed to the obesity epidemic, given the strong association between overweight and estrogen receptor-positive cancer [36,37]. According to Rahib L et al, breast cancer is projected to remain the first female cancer by 2040 in US [38]. The decline in breast cancer incidence rates has also been demonstrated using machine learning prediction models in the European continent. In fact, when the incidence rates of 2012 were considered, only two countries, Switzerland and Italy had increasing incidence rates for Breast Cancer in 2020 (1.0% and 1.8% respectively). However, fluctuations occurred for Austria during the period between 2014 and 2019. All other countries have stable reductions of incidence rates during this period [39].

**Table 3. Projection of female breast cancer crude rates (/$10^5$ person-years) by age group until to 2030 in the Monastir region.**

| Year | 20–39 | | | 40–59 | | | ≥ 60 | | |
|---|---|---|---|---|---|---|---|---|---|
| | CIR* | 95% CI | | CIR* | 95% CI | | CIR* | 95% CI | |
| | | Lower limit | Upper limit | | Lower limit | Upper limit | | Lower limit | Upper limit |
| 2016 | 22.15 | 15.61 | 31.42 | 143.36 | 117.27 | 175.23 | 151.29 | 112.50 | 203.40 |
| 2020 | 24.59 | 14.93 | 40.46 | 174.06 | 130.01 | 233.04 | 183.17 | 118.70 | 282.64 |
| 2025 | 27.93 | 14.01 | 55.68 | 222.06 | 147.49 | 334.35 | 235.54 | 127.74 | 434.28 |
| 2030 | 31.82 | 13.15 | 77.01 | 283.05 | 166.90 | 480.01 | 301.07 | 136.34 | 664.84 |

*CIR: Crude Incidence Rate (/$10^5$ person-years).

With regard to age group, our results showed that the highest increase was shown for women aged more than 60 years. Findings from other developing countries like Pakistan showed large increases in breast cancer rates among women aged 50 to 64 years [40]. Lima SM et al analyzed trends of breast cancer incidence in 182 countries in the world between 1990 and 2017. Their results showed that the largest increase in incidence rate was in women under 50 years. Among this age group, the APC was of 1.55% worldwide and the largest increase rate was in Middle East and North Africa with 2.63% of APC [25].

In addition to the high increasing incidence of female breast cancer, this work highlights the young age at diagnosis. Indeed the median age was 49 years (IQR: 41–59 years) which is close to previous findings from Tunisia [15,41] and from other Arabic countries, such as Emirates, Oman, and Qatar [42]. The median age reported here was younger than that described in most developed countries such as in the US (median age: 61 years) [43]. In our study, about 10.5% of breast cancer cases occurred in women under 35 years old. In other Tunisian studies, this percentage ranged from 6.7% to 17% [15,16,41]. Whereas, in western countries the majority of breast cancer cases occurs in women aged more than 50 years [3,20,43]. This result was in accordance with many reports from developing countries sharing the fact that breast cancer occurs in a younger population as compared to the west which can be explained partially by the young age pyramid in LMICs [44]. Knowing that breast cancer in young female tends to be more aggressive and may be associated with an increased risk for contralateral breast cancer, this age group should have a special attention to improve prevention, diagnosis and prognosis [45].

## Limitations

To the best of our knowledge this is the first study in Monastir describing the incidence and chronological trends of female breast cancer over a period of 12 years. Nevertheless, it had some limitations. Indeed, our analysis used demographic data from the 2004 and 2014 Tunisian National Institute of Statistics Census and data on the population estimate in other years were given through projections. Another limitation is that a selection bias may be observed due to the lack of notification of private sector cancer new cases. That would slightly underestimate the real cancer incidence in Monastir governorate. Additionally, mortality which is an important epidemiological indicator of disease severity was not studied due to the lack of data.

## Conclusion

In conclusion, breast cancer had a relatively high incidence in Monastir comparing to other Tunisian regions with an earlier age at diagnosis and increasing trends over time. Thus, this disease is of a serious concern in Tunisia particularly in Monastir. Strengthening preventive measures is a primary step to hold this burden. It includes the revision of the current screening and management strategies of breast cancer. Moreover, this study underlines the importance of focusing on early detection of breast cancer. Also, intensifying public awareness campaigns and improving health care delivery system could reduce the increasing trend of the disease and its socio-economic burden. Moreover, the control of breast cancer would provide a great opportunity for further reduction of cervical cancer as well as the control of other cancers and non-communicable diseases known to share similar risk factors with breast cancer.

## Supporting information

**S1 File. Female breast cancer new cases in Monastir between 2002 and 2013.**
(XLSX)

## Acknowledgments

The authors wish to thank all the staff of the cancer register for their help in data collection. And we take this opportunity to extend our gratitude to all the team of the Department of Epidemiology and Preventive Medicine at Monastir University for their implication in data collection and analysis.

## Declarations

### Ethical approval and consent to participate

The study protocol was approved by the Ethical Committee of faculty of medicine (Monastir). As this was a retrospective study and the anonymity was respected, the need for consent deemed unnecessary according to ethical Committee of faculty of medicine (Monastir).

All methods were carried out in accordance with relevant guidelines and regulations.

## Author Contributions

**Conceptualization:** Imen Zemni, Asma Sriha Belguith.

**Formal analysis:** Imen Zemni, Meriem Kacem, Wafa Dhouib, Cyrine Bennasrallah, Hela Abroug, Manel Ben Fredj, Asma Sriha Belguith.

**Methodology:** Imen Zemni.

**Supervision:** Rim Hadhri, Moncef Mokni, Asma Sriha Belguith.

**Validation:** Ines Bouanene, Asma Sriha Belguith.

**Writing – original draft:** Imen Zemni, Meriem Kacem.

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
