## [Decision Letter · Decision Letter 0]

21 Jan 2022

PONE-D-21-39057Breast cancer incidence and predictions (Monastir, Tunisia: 2002-2030): A registry-based studyPLOS ONE

Dear Dr ZEMNI Imen,

Warm regards,

On behalf of PLOS One  Team , I Thank you for submitting your manuscript to PLOS ONE. After careful consideration, we feel that it has merit but does not fully meet PLOS ONE’s publication criteria as it currently stands. Therefore, we invite you to submit a revised version of the manuscript that addresses the  minor reveiwer points raised during the process.

At the outset, I would congratulate authors to highlight the Breast Cancer trends in context to Monastir, Tunisia. This would further enrich the scientific community on the topic. It is kindly conveyed that, to increase the scientific validity of manuscript please add/clarify/incorporate  the points raised by reviewer, e.g Denominator population, discussion articles etc.Please revisit the manuscript for few typos also  Authors had highlighted one of the key observation, " In our study, about 10.5% of breast cancer cases occurred in women under 35 years old" which would raise the concern of policy makers for strengthening early screening and diagnosis.  Recently, across the world we had observed similar trend, particularly in  young females as reported in the submitted manuscript also . Study enforces the need for appropriate interventions for early screening and diagnosis. ==============================

PLOS ONE team  looks forward to receiving your revised manuscript at earliest feasible time.

Kind regards,

Gopal Ashish Sharma, MBBS, MD

Academic Editor

PLOS ONE

Journal Requirements:

Reviewers' comments:

Reviewer's Responses to Questions

**Comments to the Author**

1. Is the manuscript technically sound, and do the data support the conclusions?

Reviewer #1: Yes

Reviewer #2: Yes

2. Has the statistical analysis been performed appropriately and rigorously? 

Reviewer #1: Yes

Reviewer #2: Yes

3. Have the authors made all data underlying the findings in their manuscript fully available?

Reviewer #1: Yes

Reviewer #2: Yes

4. Is the manuscript presented in an intelligible fashion and written in standard English?

Reviewer #1: Yes

Reviewer #2: No

5. Review Comments to the Author

Reviewer #1: 1. Abstract: The second line "Among females'.....causing mortality" needs rephrasing.

2. For the worldwide readers, it would be better that the spelling of "Crud" as in crude incidence rate be written uniformly as "Crude"

3. In methodology section, As you have included only females with new breast cancer in your study, (Male breast cancer cases were excluded), it is not clear that you used sex specific population as denominator or not. The title of the relevant tables should also mention the word "females".

4. The time point of the popuation used as denominator (eg mid-year...or something else ) is not mentioned.

5. Statistial analysis section should mention the full details of the SPSS software used like the city, country etc.

6. In discussion section, Spelling of "contralateral" (Line with refefrence no. 30)

Reviewer #2: This is an ambitious endeavour the authors undertook and they have been successful in reaching a conclusion that is useful for both the scientific community and general population alike.

The paper follows IMRAD. Introduction well enunciates the background of the problem and reason for its exploration.

The methodology is adequate but I could not locate the ethics committee recommendations or consent.

The results are lucid and in agreement with the latest GLOBOCAN estimates.

The discussion in this paper is a momentous tasks and needs more trends to be discussed worldwide, although the authors have tried to reason well that why Montasir has higher breast cancer incidence rate in comparison to

the national rate and how does it stand globally.

May I suggest more scientific discussions to the authors.

You are also requested to compare with predictions from Machine Learning Models.

https://acsjournals.onlinelibrary.wiley.com/doi/full/10.3322/caac.21660

https://journals.sagepub.com/doi/full/10.1177/1460458220983878

https://www.ncbi.nlm.nih.gov/pmc/articles/PMC6659231/

https://jamanetwork.com/journals/jamanetworkopen/fullarticle/2778204

https://www.thelancet.com/journals/eclinm/article/PIIS2589-5370(21)00265-0/fulltext

chrome-extension://efaidnbmnnnibpcajpcglclefindmkaj/viewer.html?pdfurl=http%3A%2F%2Fjournal.waocp.org/article_16333_aa05a4f1e68d4a4a60d9936cac9a2ad2.pdf

https://www.karger.com/Article/FullText/503219

https://www.koreascience.or.kr/article/JAKO201525249375821.page

There were no limitations and biases mentioned in the paper and authors are requested to add them as well.

spelling errors noted (crud = crude)

6. PLOS authors have the option to publish the peer review history of their article (what does this mean?). If published, this will include your full peer review and any attached files.

Reviewer #1: **Yes: **Dr Vijay Kumar Barwal

Reviewer #2: **Yes: **Vidisha Vallabh

---

## [Author Response · Author response to Decision Letter 0]

1 Apr 2022

Response to Reviewers

Dear Editor,

Thank you for your attention to our work. The comments and suggestions we have received are valuable and very helpful to improve our manuscript. We have made revisions based on your latest comments and suggestions, as described in the authors' response. 

We respond in detail to each of the reviewers’ comments. They raised important issues and we agree with almost all their comments. We have revised our manuscript according to their indications. We hope that they will find our responses to their comments satisfactory, and we are willing to finish the revised version of the manuscript including any further suggestion that the reviewers may have. The revision has been developed in consultation with all coauthors, and each author has given approval to the final form of this revision. All changes made in the revised version will be visible in red.

 Please find as attached files: 

The “Response to Reviewers”, 

the “'Revised Manuscript with Track Change” and 

the “Manuscript” (the unmarked version of the revised manuscript). 

We sincerely hope that the revision of the manuscript will be satisfactory and the enclosed version will be acceptable for publication. 

Once again thank you for your cooperation.

Sincerely Yours,

Dr Imen ZEMNI

Journal Requirements:

Response: Thank you. We did the required modifications.

2. In your Data Availability statement, you have not specified where the minimal data set underlying the results described in your manuscript can be found. 

Response: Thank you. We attached an Excel supplementary file intitled “S1 File. Female breast cancer new cases in Monastir between 2002 and 2013” in Supporting Information. (Lines 461 and 462)

3. PLOS requires an ORCID iD for the corresponding author in Editorial Manager on papers submitted after December 6th, 2016. Please ensure that you have an ORCID iD and that it is validated in Editorial Manager. 

Response: Thank you. We indicated the ORCID iD of the corresponding author.

4. Please review your reference list to ensure that it is complete and correct.

Response: Thank you. We reviewed the reference list and added other references as it was recommended by reviewer 2.

Review Comments:

Reviewer 1:

1. Abstract: The second line "Among females'.....causing mortality" needs rephrasing.

Response: Thank you. We did the necessary modifications. (Lines 32 and 33)

2. For the worldwide readers, it would be better that the spelling of "Crud" as in crude incidence rate be written uniformly as "Crude".

Response: Thank you. We did the necessary modifications. (Lines 82,113,141,142,145,149 and 305)

3. In methodology section, As you have included only females with new breast cancer in your study, (Male breast cancer cases were excluded), it is not clear that you used sex specific population as denominator or not. The title of the relevant tables should also mention the word "females".

Response: Thank you. As we have included only females with new breast cancer in our study, we used female population as denominator. We added this information in the methodology section (Lines 115 and 116). We also mentioned the word "female" in the titles of the relevant tables (Lines 155,161 and 179).

4. The time point of the population used as denominator (eg mid-year...or something else ) is not mentioned.

Response: Thank you. We computed the point incidence rate of breast cancer (per 100,000 population) for each year based on the Tunisian National Institute of Statistics data. Incidence rates were expressed per 100,000 person-years: The Tunisian National Institute of Statistics carries out censuses of the Tunisian population every 10 years. The last censuses were carried out between April and May 2014. Data on the population estimate in other years are given through projections made by the Tunisian National Institute of Statistics.

 5. Statistical analysis section should mention the full details of the SPSS software used like the city, country etc.

Response: Thank you. We did the necessary modifications. (Lines 112 and 113)

6. In discussion section, Spelling of "contralateral" 

Response: Thank you. We corrected the spelling. (Line 276)

Reviewer 2:

1. The methodology is adequate, but I could not locate the ethics committee recommendations or consent.

Response: Thank you. We mentioned in the methodology section that the protocol of this study was approved by the ethics Committee of the faculty of medicine of Monastir (Reference Number: IORG0009738N°101/OMB0990-0279).) (Lines 124,125 and 126)

2. The discussion in this paper is a momentous tasks and needs more trends to be discussed worldwide, although the authors have tried to reason well that why Monastir has higher breast cancer incidence rate in comparison to the national rate and how does it stand globally. May I suggest more scientific discussions to the authors. You are also requested to compare with predictions from Machine Learning Models.

Response: Thank you. We revised the discussion and we added paragraphs to discuss:

The incidence rate in Monastir comparatively to other national results and international findings (lines 201-211).

Trends of breast cancer incidence comparatively to last reports from Tunisia and to statistics from the different regions in the World (Lines 230-257).

Trends according age groups (Line 258-267).

3. There were no limitations and biases mentioned in the paper and authors are requested to add them as well.

Response: Thank you. We added a paragraph for limitations at the end of the manuscript (Lines 279-287).

4. Spelling errors noted (crud = crude)

Response: Thank you. We did the necessary modifications. (Lines 82,112,136,137,144 and 264)

---

## [Decision Letter · Decision Letter 1]

21 Apr 2022

Breast cancer incidence and predictions (Monastir, Tunisia: 2002-2030): A registry-based study

PONE-D-21-39057R1

Dear Dr.Imen Zemni,

We’re pleased to inform you that your manuscript has been judged scientifically suitable for publication and will be formally accepted for publication once it meets all outstanding technical requirements.

Kind regards,

Gopal Ashish Sharma, MBBS, MD

Academic Editor

PLOS ONE

Additional Editor Comments (optional):

Revised manuscript is accepted for publication as both the reviewers had conveyed the accepatnce of changes proposed through journal and available electronic means. 

Reviewers' comments:

Reviewer's Responses to Questions

**Comments to the Author**

1. If the authors have adequately addressed your comments raised in a previous round of review and you feel that this manuscript is now acceptable for publication, you may indicate that here to bypass the “Comments to the Author” section, enter your conflict of interest statement in the “Confidential to Editor” section, and submit your "Accept" recommendation.

Reviewer #1: All comments have been addressed

2. Is the manuscript technically sound, and do the data support the conclusions?

Reviewer #1: Yes

3. Has the statistical analysis been performed appropriately and rigorously? 

Reviewer #1: Yes

4. Have the authors made all data underlying the findings in their manuscript fully available?

Reviewer #1: Yes

5. Is the manuscript presented in an intelligible fashion and written in standard English?

Reviewer #1: Yes

6. Review Comments to the Author

Reviewer #1: I thanks all the authors for considering our suggestions regarding the revision and further improvement of this manuscript.

7. PLOS authors have the option to publish the peer review history of their article (what does this mean?). If published, this will include your full peer review and any attached files.

Reviewer #1: **Yes: **Dr. Vijay Kumar Barwal, IGMC Shimla, India

---

## [Editor Report · Acceptance letter]

16 May 2022

PONE-D-21-39057R1 

Breast cancer incidence and predictions (Monastir, Tunisia: 2002-2030): A registry-based study 

Dear Dr. Zemni:

I'm pleased to inform you that your manuscript has been deemed suitable for publication in PLOS ONE. Congratulations! Your manuscript is now with our production department. 

Kind regards, 

on behalf of

Dr. Gopal Ashish Sharma 

Academic Editor

PLOS ONE